# Cognitive stimulation in activities of daily living for individuals with mild-to-moderate dementia (CS-ADL): Study protocol for a randomised controlled trial

**Simone M. Ryan** [1][ORCID][☺], **Orla Brady** [1,2][ORCID][☺] *

**1** Discipline of Occupational Therapy, School of Health Sciences, Áras Moyola, University of Galway, Galway, Ireland, **2** Primary Care, HSE, Trim, Meath, Ireland

☺ These authors contributed equally to this work.
* s.ryan72@universityofgalway.ie

**Data Availability Statement:** This study will adhere to the General Data Protection Regulation (GDPR), the Irish Data Protection Act (2018) and specific institutional data protection and management

## Abstract

### Background

Multi-component CS programs incorporating practice of activities of daily living (ADL) into intervention have reported benefits for ADL outcomes in individuals living with mild-to-moderate dementia. A randomised controlled trial (RCT) within community occupational therapy services in Ireland, is planned to evaluate the effects of CS-ADL, an ADL-focused, multi-component CS program, on ADL outcomes for individuals living with mild-to-moderate dementia.

### Method

A single-blind RCT with a calculated sample size of 34 participants has been planned to compare the effects of CS-ADL versus treatment as usual on the outcomes of basic ADLs and instrumental ADLs. Cognition, mood, communication, and quality of life will also be evaluated as secondary outcomes. CS-ADL sessions will run once weekly for a total of seven weeks, lasting approximately two hours each. Outcome data will be collected at baseline, within sessions and post-intervention at week eight. Descriptive statistics will be used to analyse the data. This study has been registered at clinicaltrials.gov (NCT06147479).

### Discussion

CS programs are commonly conducted by occupational therapists working with individuals living with mild-to-moderate dementia. This study aims to demonstrate the effectiveness of a multi-component CS program delivered through an occupational therapy lens, potentially influencing the approach to CS and ADL interventions undertaken by occupational therapists.

policies for the University of Galway. Any personal, identifiable information will be stored securely in a locked cupboard in the occupational therapy department of the University of Galway, and only accessible to the principal investigator and their academic supervisors. This information is securely destroyed following completion of the study. As this article is a study protocol, no datasets have currently been generated or analysed. All relevant, anonymised data will be made publicly available upon study completion via an online repository (e.g., Figshare), with study results disseminated through academic outputs such as peer-reviewed publications and conference presentations.

**Funding:** The author(s) received no specific funding for this work.

**Competing interests:** The authors have declared that no competing interests exist.

## Introduction

Dementia is a clinical syndrome characterised by deteriorating cognitive, behavioural and emotional functions that interfere with an individual's performance in activities of daily living (ADL), [1, 2]. The cognitive and social changes associated with the condition have been linked to decreased quality of life, increased caregiver burden, and higher care costs, contributing to dementia being one of the leading causes of hospitalisations and admissions to skilled nursing facilities worldwide [3]. Dementia affects approximately 57.4 million people globally, with an estimated 53,932 individuals in Ireland living with dementia as of 2019 [4, 5]. The widespread impact of dementia has prompted the development of a variety of treatments to slow the cognitive and functional decline associated with this condition [6]. Several non-pharmacological interventions are increasingly being used to mitigate this decline, with cognitive stimulation (CS) being one such approach.

CS is a non-pharmacological approach to intervention widely used with individuals with early-stage dementia. CS is typically delivered in a stimulating and rewarding environment, where participants engage in a range of activities and discussions aimed at enhancing global cognitive and social functioning, as opposed to a focus on specific functions in isolation [7]. A variety of CS interventions have been described across the literature, including manualised therapies, individual programs, and CS delivered as part of a wider, multi-component intervention [8]. Cognitive stimulation therapy (CST) is one notable example of a manualised therapy commonly used by occupational therapists for people living with mild-to-moderate dementia. CST is a brief group intervention based on CS principles, consisting of 14 sessions of themed activities and discussions run over seven weeks [9]. Several adaptations of this CST program have been developed and evaluated in the literature, including a 24-week maintenance program (MCST) and an individualised program (iCST). CST has demonstrated significant benefits for cognition, quality of life and communication for individuals with mild-to-moderate dementia, as reported by Woods et al.'s (2012) systematic review [10]. While it is assumed that cognitive improvements following CS interventions will transfer to everyday functioning, evidence from the literature does not support this assumption; no statistically significant effects of CS for ADL performance were reported [10].

However, Ryan & Brady's (2023) scoping review [8] exploring the use of CS in improving ADL outcomes for individuals with mild-to-moderate dementia mapped a variety of alternative CS programs that reported benefits for ADLs. Three randomised controlled trials (RCTs) with an overall total of 252 participants evaluated the effect of a German, multi-component group therapy termed MAKS [11–13]. MAKS sessions consist of physical activities, individual/group cognitive stimulation exercises, and participation in ADL tasks, with all three studies reporting benefits of MAKS on ADL abilities. A cognitive-motor stimulation intervention (CMSI) was evaluated at 12 months by [14] and at three years by [15] through an RCT design. CMSI sessions consisted of individual/group cognitive exercises, psychomotor therapy, and ADL training. Favourable ADL outcomes were reported for the CMSI group when compared to a control. Furthermore, Jiménez Palomares et al.'s (2021) pilot RCT ($N = 58$) [16] investigating the effects of an occupational therapy CS program on ADL skills reported significant improvements in basic ADLs such as feeding, dressing and continence. Ryan and Brady's (2023) scoping review [8] observed a common trend across studies reporting positive effects for ADLs; multi-component interventions incorporating CS with ADL practice and/or physical activity produced more favourable ADL outcomes in comparison to traditional, discussion-based CS programs, like CST.

Occupational therapy is a profession where occupational performance and engagement in ADLs are central to intervention. While the nature of the scoping review methodology limits

conclusions that can be drawn, the reduced evidence supporting the benefits of traditional CS programs for ADLs calls for a shift in approach used by occupational therapists [8]. The incorporation of ADL-focused activities into multi-component CS programs demonstrate promising results for ADL outcomes [11, 15]. However, the majority of multi-component CS interventions reported to benefit ADL outcomes by Ryan and Brady (2023) [8] recruited participants from nursing homes or long-term care facilities. The lack of data evaluating the implementation of these programs within the community setting prompted the development of CS-ADL, a multi-component group CS program that aims to enhance ADL performance alongside social and cognitive functioning.

CS-ADL is considered a complex intervention due to the number of components, the range of behaviours targeted, and the level of expertise required of those delivering the intervention [17]. Development of the CS-ADL program thus far has aligned with the Medical Research Council (MRC) guidance for developing and evaluating complex interventions [18]. The CS-ADL program was scientifically developed based on the currently available literature evidence and appropriately revised after it was successfully piloted in an Irish, community-based Psychiatry of Later Life service. Subsequently, a qualitative investigation of the participant and caregiver experience of CS-ADL identified that both participants and their caregivers perceive CS-ADL to be an acceptable intervention that positively influenced the daily lives of both dyad members. Notably, benefits were reported in the memory, mood, and social interaction of participants [17]. This research engaged key stakeholders to provide a preliminary evaluation of the effect and feasibility of CS-ADL, with findings used to further refine the intervention, in line with MRC guidance. The development of complex interventions is an iterative process; phases often overlap or are repeated to resolve uncertainties during development. Future research is required to economically evaluate the program, and to further evaluate the process, feasibility and implementation of CS-ADL in clinical practice. Additionally, the effectiveness of CS-ADL on improving ADL outcomes must be explored. Therefore, an RCT evaluating the effect of CS-ADL on ADL outcomes when compared to a group receiving treatment as usual (TAU) is proposed. This RCT will take place in Irish community occupational therapy services, and individuals living with mild-to-moderate dementia will be recruited to take part.

## Aims and objectives

The aim of this study is to evaluate the effect of CS-ADL, an ADL-focused multi-component CS program, on ADL outcomes for people living with mild-to-moderate dementia. The primary objective of this study is to determine the effect of CS-ADL on basic ADL (BADL) and instrumental ADL outcomes (IADL). The secondary objectives of the study include examining the effects of the intervention on cognition, mood, quality of life (QOL) and communication. These secondary outcomes were chosen as previous CS programs have demonstrated benefits for these variables [10].

## Materials and methods

### Trial design

This will be a multi-centre, parallel, single-blind, randomised trial of CS-ADL versus TAU for people with mild-to-moderate dementia. Sample size was calculated using formal power analysis with G*Power software [19]. Based on Cohen's (1988) guidelines [20] for small ($r = 0.1$), medium ($r = 0.3$) and large ($r = 0.5$) effects for a 2 (pre- and post-test) x 2 (CS-ADL and TAU) $t$-test, with the difference between two dependent means (matched pairs), sample size was calculated for an estimated small ($d = 0.2$), medium ($d = 0.5$) and large ($d = 0.8$) effect size. Two-tailed alpha of 0.05 was assumed for all tests. With an estimated effect size of 0.2, a sample size

of 199 is required for a critical t of 1.9720175 and degrees of freedom (df) of 198, with an actual power of 0.8016910. With an estimated effect size of 0.5, a sample size of 34 is required for a critical t of 2.0345153 and a df of 33 and an actual power of 0.8077775. For an estimated effect size of 0.8, a sample size of 15 was calculated to be required for a critical t of 2.1447867 and a df of 14, with an actual power of 0.8213105. It is expected there will be challenges in the recruitment of 199 participants for the estimated effect size of 0.2, due to the scale and timeframe of this study (approximately 1 year). Furthermore, the challenges recruiting individuals with dementia to randomised controlled trials has been widely noted across the literature [21]. Therefore, for pragmatic reasons, recruitment will aim for the sample size of 34 (*d* = 0.5). Participants will be randomised with 1:1 allocation ratio to either the intervention or control group. The primary and secondary outcomes will be measured at baseline prior to randomisation and at the end of intervention (8 weeks). There will also be an observational assessment of occupational performance completed by the treating therapists within every group session. While this study is of a short duration, it is anticipated that there will be a minimum 10% attrition rate. This figure stems from an analysis of dementia trial retention data [22]. Therefore, recruitment will aim to exceed the estimated sample size by 10% to ensure the study is adequately powered.

## Study setting

The study will take place in Irish Health Service Executive (HSE), community-based occupational therapy services. The principal investigator will deliver the CS-ADL intervention.

## Eligibility criteria

Eligibility criteria for this study reflects criteria of previous multi-component CS interventions. Participants must have a formal diagnosis of major neurocognitive disorder (dementia) as per the DSM-V criteria [1]. Participants must have a mild-to-moderate cognitive impairment as classified by the Mini Mental State Examination (MMSE) or equivalent screens [23]. Individuals with severe dementia, i.e., an MMSE<10, will be excluded as most CS interventions are not applicable for those with severe cognitive impairment [9]. Participants taking dementia medication can continue to do so during the study.

Participants must have some ability to communicate and understand communication, determined by a score of 1 or 0 on questions 12 and 13 of the Clifton Assessment Procedures for the Elderly-Behaviour Rating Scale [24]. Participants must be able to see and hear well enough to participate in the group. Participants will be excluded if they have significant uncontrolled disruptive behaviours, a premorbid diagnosis of a learning disability, or a significant physical illness/disability that may affect participation during intervention sessions or assessments.

## Recruitment

Participants will be recruited from HSE occupational therapy settings in Ireland. Clinicians will be encouraged to screen their case load for potential participants and approach them and their caregiver to discuss the study. If they are interested in taking part, internal referral protocols will be followed. Contact details will be passed to the research team who will arrange a meeting with the participant and their caregiver to assess eligibility. If they are eligible and agree to take part, informed written consent will be obtained. Participants' capacity to consent to take part in the study will be assessed following the guidelines stipulated in the Assisted Decision-Making (Capacity) Act (2015) [25]. If the participant with dementia lacks the capacity to consent, appropriate protocols will be adhered to in line with the Assisted Decision-

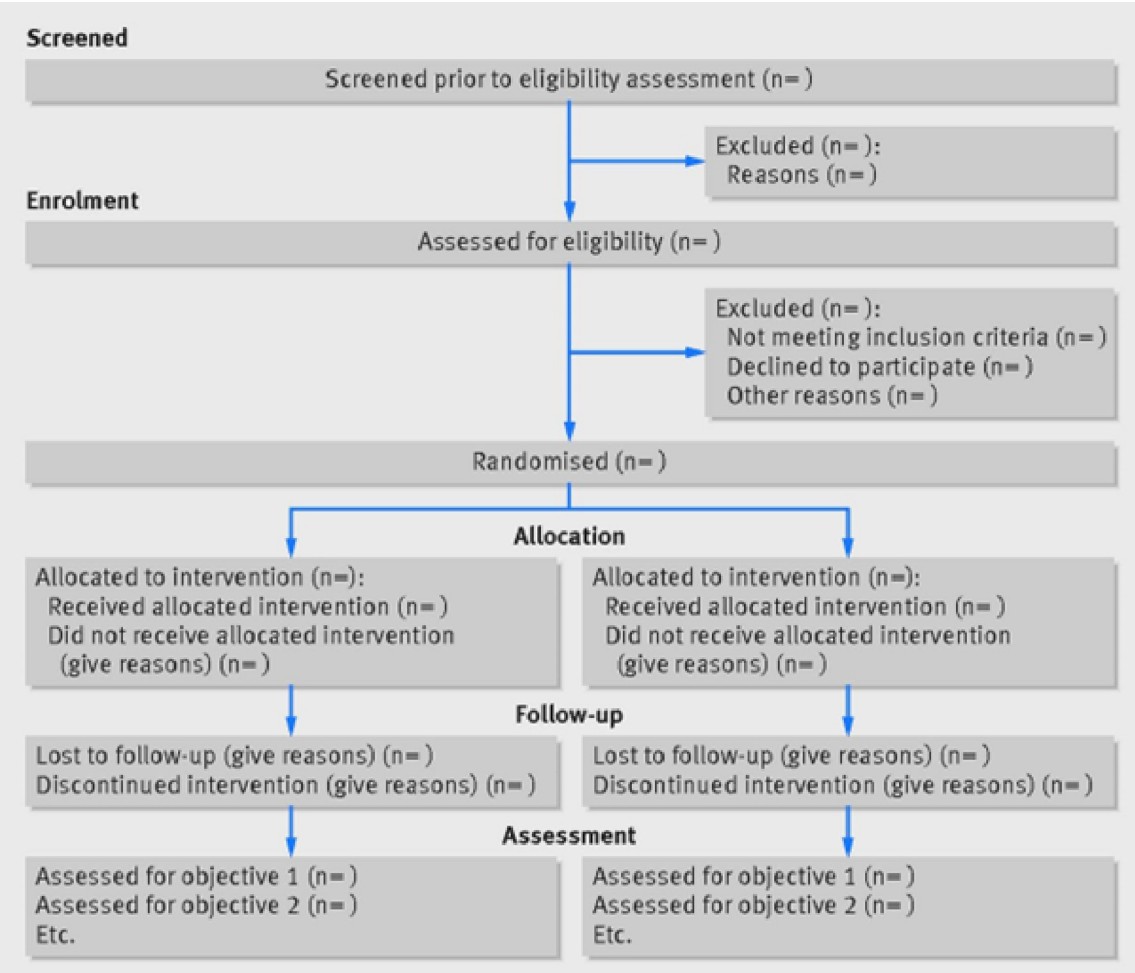

**Fig 1. Participant flow diagram.**

Making (Capacity) Act (2015) which has recently evolved into full enactment in Ireland [25]. Fig 1 displays the planned participant timeline throughout the study.

## Randomisation and masking

Randomisation will occur after eligibility, consent and baseline assessments have been carried out. Random allocation will be completed by one of the principal investigator's academic supervisors who are otherwise uninvolved in the conduct of the study. A computer-generated number system (RANDBETWEEN Command on Microsoft Excel) will be used. While participants cannot be blinded to their allocated group, outcome assessors will be blind to group allocation. While it is possible that unblinding might occur, i.e., outcome assessors become aware of group allocation, the process of blinding will be explained to participants, and they will be asked to not reveal their treatment allocation to the assessor. The success of blinding will be tested following completion of outcome measures by asking assessors to guess the treatment assignment of participants, expressing either a treatment guess or uncertainty. If blinding is successful, 'do not know' guesses should be high, and incorrect guesses should be balanced [26]. However, as some outcome measures are caregiver-rated, it will not be possible to blind caregivers to group allocation.

## Intervention

**CS-ADL.** The intervention group will receive a program of CS-ADL, with sessions delivered once weekly across 7 weeks. Typical CST programs commonly delivered by Irish occupational therapy services span a duration of 7 weeks, demonstrating that an intervention of this length is feasible within these services. CS-ADL sessions will last approximately 2 hours. While significant ADL benefits from intensive multi-component CS interventions [11, 15] have been reported [8], an occupational therapy CS program reported benefits for ADLs through a less intensive approach of two 45-minute sessions per week for a total of 5 weeks [16]. As typical community-based CST programs often deliver two 45-minute sessions on the one day to aid feasibility, the planned intensity and duration of the CS-ADL program is therefore justified. At least one qualified occupational therapist will facilitate the sessions, with the assistance of a co-facilitator, as CS-ADL activities require a level of supervision not feasible for a single facilitator. Furthermore, the lead group facilitator must be an occupational therapist as the aims of CS-ADL align with the core principles of the profession. Each session consists of similar components, however the sequence and timing allocated per component may differ between sessions. Activities are based around a different theme for each session, for example morning routine, evening routine, or domestic activities, with each session intended to be enjoyable and mentally stimulating for the individual. A typical session will begin with 20-minutes of introductions, reality orientation, a group song and discussion of the news of the week. 15-minutes will then be spent on a physical activity/game. Physical activity will be a key component of the CS-ADL program as 7 of 9 CS programs reviewed by Ryan and Brady (2023) involving a physical activity component to intervention reported benefits for ADL outcomes. A physical activity at the start of the group will also aim to benefit arousal levels of participants and build rapport amongst group members. This will be followed by 25-minutes of ADL-focused cognitive stimulation activities, a 15-minute break, and approximately 10 more minutes of ADL-focused CS activities. These activities will include the identification and categorisation of everyday items, discussion, reminiscence, planning and sequencing of ADLs and the completion of everyday writing/calculation tasks. This will be followed by a 30-minute ADL task, which is an essential component to the program, as CS interventions with an ADL component report significant benefits for ADL outcomes [8]. Examples of activities will include making a breakfast, a simple gardening activity or sorting laundry. While the structure of CS-ADL has taken inspiration from previous CS programs [9, 11, 16].CS-ADL differs as participants are actively engaged in the planning, preparation, and practice of everyday activities like washing, dressing, and cooking. Further detail of the ADL components of the intervention program are provided in the (S1 Table). Therapists involved in the administration of CS-ADL are advised to use their clinical judgement in the planning and implementation of sessions, as sessions should be individualised to suit the needs, abilities, and interests of group members. Activities should be adapted accordingly. Intervention will be delivered in therapy space available at each clinical site where recruitment takes place.

**Treatment-as-usual.** The control group will continue to have access to treatment as usual, which will include medication, input from health professionals and any activities provided in the day hospital if the participant is in attendance. After the completion of the study, participants and their caregivers will be offered a home programme based on the CS-ADL intervention and will be given training in how to use it. Treatment-as-usual was chosen as a comparator as it would be unethical to ask both the intervention and comparator group to cease treatment for the purposes of this study.

**Outcome measures.** Outcome measures will be recorded in both CS-ADL and TAU participants at baseline (week 0) and post-intervention (week 8). The completion of outcome

measures pre- and post-intervention will be mandatory for participation in the trial; participants and caregivers will be reminded of requirements prior to the commencement of the group and after the completion. Outcomes will be assessed by an assessor blind to group allocation. Socio-demographic information of participants will be collected including age, gender, ethnicity, diagnosis, and medication.

**Primary outcomes: ADL performance.** Change in ADL performance will be measured using the Alzheimer's Disease Cooperative Study-Activities of Daily Living (ADCS-ADL) scale [27]. This scale was designed for the assessment of ADL outcomes for individuals with Alzheimer's disease in clinical trials. The scale consists of 23 items that includes 6 BADL and 17 IADL outcomes, scored on a range from 0 (patient does not perform the activity), to the highest score (patient is independent in the activity). The scale is administered with caregivers who are asked to rate the degree to which their care-recipient performs each item in the last four weeks. The ADCS-ADL has been identified to be a reliable and valid instrument, with high internal consistency and sensitivity to functional changes in individuals with mild-to-moderate dementia [28].

**Secondary outcomes: Cognition, communication, QOL, mood.** General cognitive functioning will be measured using either the Montreal Cognitive Assessment (MoCA) face-to-face or the MoCA-Blind delivered via the telephone, dependent on feasibility for participants and assessors. The MoCA is a brief screening tool used to detect cognitive impairment and consists of a 30-point test assessing the cognitive domains of visuospatial/executive, naming, memory, attention, language, abstraction, delayed recall, and orientation. The MoCA exhibits high sensitivity and specificity and its validity for use with people living with dementia has been well established [29]. The MoCA-Blind does not require visual input and was developed to enable cognitive screening of those with visual impairment, also enabling administration through telephone or online formats. The MoCA-Blind is scored on a scale of 22 points as visual elements are excluded. While specificity remains on par with the original MoCA, sensitivity is reduced [30].

Cognition will be assessed using the Repeatable Battery for Assessment of Neuropsychological Status (RBANS) [31]. The RBANS is a commonly used brief battery of cognitive function for individuals with dementia. This battery consists of 12 subtests which assess five cognitive domains: immediate memory, visuospatial-constructional ability, language and delayed memory. This battery has demonstrated good reliability and adequate validity indicators.

Communication ability will be assessed using the Holden Communication Scale (HCS) [32]. This scale includes 12 items assessing the domains of conversation, awareness, humour, and responsiveness. Each items contains five response options ranging from 0 to 4. Total scores can range from 0 to 48 with higher scores indicating more communication difficulties. The HCS was initially developed to assess communication outcomes in reality orientation and reminiscence programs and has demonstrated good reliability and validity for use with people living with dementia [33]. While there are limitations to reliability and validity data outside of the original development of the tool [2], the HCS has been used in various CS studies to assess communication e.g. [9, 34].

QOL will be assessed using the Quality of Life-Alzheimer's Disease Scale (QOL-AD). This scale consists of 13 items spanning the domains of physical health, energy, mood, living situation, memory, family, marriage, friends, chores, fun, money, self, and life as a whole. Response options include 1 (poor), 2 (fair), 3 (good) and 4 (excellent), with total scores ranging from 13–52. Assessment is delivered through an interview format and separate ratings are obtained from the participant themselves and the caregiver. Higher scores indicate a better quality of life. The QOL-AD has demonstrated good reliability and internal consistency [35] and has been found to detect improvements in QOL in previous CS studies [9, 34].

Mood will be assessed through the presence of neuropsychiatric symptoms as measured by the Neuropsychiatric Inventory Questionnaire (NPI-Q), a version of the Neuropsychiatric Inventory (NPI) [36]. The NPI-Q is a caregiver-based questionnaire that measures the presence and severity of 12 neuropsychiatric symptoms in people living with dementia, including delusions, hallucinations, apathy, disinhibition, and agitation/aggression. The severity scale runs from 1 to 3 points (1 = mild, 2 = moderate, 3 = severe). The questionnaire also measures the respective level of caregiver distress associated with each symptom, with the scale running from 0 to 5 points (0 = no distress, 5 = extreme distress). The scale is widely used in research, has demonstrated acceptable levels of internal consistency, and has been cross validated with the standard NPI in clinical practice settings [37].

Outcome assessors will be blinded to group allocation and will have received adequate training and instruction in their administration. Participants will be informed that participating in both pre- and post-intervention outcome assessments is a requirement for involvement in the study.

**Other outcome measures.**   Occupational performance within a group setting will be measured after each session using the Occupational Therapy Task Observation Scale (OTTOS). The OTTOS contains two parts, with 10 items for evaluation of specific task functions and 5 items for rating general behaviour. The reliability and validity of the OTTOS has been demonstrated [38].

## Statistical analysis

The chosen statistical method will depend on whether an adequate number of participants are recruited to power the study, however, this protocol will outline the planned statistical analysis method.

The study sample will be analysed using descriptive statistics through the latest version of Statistical Package for Social Sciences (SPSS). Data on recruitment will be recorded and examined. A Consolidated Standards of Reporting Trials (CONSORT) diagram describing the flow of participants through the study will be presented in the final RCT report, detailing enrolment, allocation, follow-up, and analysis of data (Fig 1) [39]. Baseline data will be assessed for normality and if normally distributed, parametric analysis will be used. Where data is not normally distributed, non-parametric analysis will be used. Types of tests chosen will be based on whether the data is parametric or non-parametric. Significance levels will be set to 0.05. A between-group analysis will be completed to examine if the groups are significantly different at outcome. A within-group analysis will be also completed to evaluate if there are any significant changes within groups. Sociodemographic factors like age and gender will be entered as covariates. It is possible that some data will be missing. In this instance, attempts to follow-up with all participants will be made. If missing data remain, intention-to-treat analysis will be used to account for this data.

Due to the proposed small sample size previously discussed for this study, and the corresponding medium effect size, the minimum clinically important difference (MCID) will be calculated. The MCID is calculated based on the standardised response mean, obtained by dividing the difference in scores from baseline to posttreatment by the standard deviation of baseline scores [40].

## Ethical considerations and trial registration

Ethical approval has been received from the relevant committee on the 27[th] September 2023 (HSE North East Research Ethics Committee; reference number REC/23/042). This study conforms to the Declaration of Helsinki and has been reviewed and registered with Clinicaltrials.

gov (NCT06147479). Changes made to the protocol throughout the conduct of the study will be logged in the trial registry.

## Anticipated risks for trial participants

There appear to be no documented harmful side effects from participating in typical CS programs, with no adverse reactions apparent. Health and safety risks, including the risk of falls may be posed by the participation in cooking tasks and physical activities. However, group facilitators will use their clinical judgement, clinical risk assessments in the implementation of activities and always ensure adequate supervision is given. Nevertheless, potential participants and their caregivers will be fully informed of the potential risks before partaking in the trial.

## Study status

This study is currently ongoing and actively recruiting participants. Recruitment commenced the 23rd of October 2023.

## Discussion

Dementia is a progressive syndrome where deteriorating cognitive, behavioural, and emotional functions impede on the individual's ability to engage in meaningful occupations and ADLs. Traditional discussion-based CS programs have not demonstrated consistent benefits for ADL outcomes for individuals with mild-to-moderate dementia. This has prompted the development of CS-ADL, an ADL-focused multi-component CS program, informed by the results of Ryan and Brady's (2023) scoping review [8] and a pilot study conducted within the Mullingar PLL services. A single-blind, parallel, randomised superiority trial has been proposed, aiming to evaluate the benefits of CS-ADL on the ADL outcomes of individuals living with mild-to-moderate dementia, in comparison to TAU. The proposed RCT also intends to evaluate the effects of CS-ADL on cognition, communication, QOL and mood of participants in comparison to TAU. To the extent of the authors knowledge, this study will be the first of its kind conducted within the Irish PLL services and will provide valuable evidence of the effectiveness and feasibility of implementation of a multi-component CS intervention in a community setting. This study aims to influence the approach to CS intervention undertaken by Irish occupational therapists working in community PLL services. The authors intend to disseminate the result of this study through publication in peer-reviewed journals, presentation at conferences and to relevant healthcare professionals, and results will be supplied to participants if desired.

## Supporting information

**S1 Checklist. SPIRIT 2013 checklist: Recommended items to address in a clinical trial protocol and related documents\*.**
(DOC)

**S1 Table. ADL components of intervention.**
(DOCX)

**S1 File.**
(DOCX)

## Author Contributions

**Conceptualization:** Orla Brady.

**Investigation:** Simone M. Ryan.

**Methodology:** Orla Brady.

**Supervision:** Orla Brady.

**Visualization:** Orla Brady.

**Writing – original draft:** Simone M. Ryan.

**Writing – review & editing:** Simone M. Ryan, Orla Brady.

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
