## [Decision Letter · Decision Letter 0]

9 Apr 2024

PONE-D-23-41253Cognitive stimulation in activities of daily living for individuals with mild-to-moderate dementia (CS-ADL): study protocol for a randomised controlled trial.PLOS ONE

Dear Dr. Ryan,

Thank you for submitting your manuscript to PLOS ONE. After careful consideration, we feel that it has merit but does not fully meet PLOS ONE’s publication criteria as it currently stands. Therefore, we invite you to submit a revised version of the manuscript that addresses the points raised during the review process. One reviewer recommends to apply the MRC framework for the development and evaluation of complex interventions and I would encourage the authors to adhere to this recommendation. It will definitely improve the protocol and should be helpful to successfully conduct the trial.

We look forward to receiving your revised manuscript.

Kind regards,

Sascha Köpke

Academic Editor

PLOS ONE

Reviewers' comments:

Reviewer's Responses to Questions

**Comments to the Author**

1. Does the manuscript provide a valid rationale for the proposed study, with clearly identified and justified research questions?

Reviewer #1: Yes

Reviewer #2: Yes

2. Is the protocol technically sound and planned in a manner that will lead to a meaningful outcome and allow testing the stated hypotheses?

Reviewer #1: Partly

Reviewer #2: Partly

3. Is the methodology feasible and described in sufficient detail to allow the work to be replicable?

Reviewer #1: Yes

Reviewer #2: No

4. Have the authors described where all data underlying the findings will be made available when the study is complete?

Reviewer #1: Yes

Reviewer #2: Yes

5. Is the manuscript presented in an intelligible fashion and written in standard English?

Reviewer #1: Yes

Reviewer #2: Yes

6. Review Comments to the Author

You may also provide optional suggestions and comments to authors that they might find helpful in planning their study.

Reviewer #1: The paper is well written. I have only a few concerns.

1. Why a medium effect size (d=0.5) is chosen, please explain the rationale behind it.

2. Two primary outcome is chosen BADL and IADL. This will require multiple testing adjustment s otherwise type-1 error will be inflated. This is not addressed.

3. How the missing data will be handled?

4. The power analysis is based on two-sample t-tests, but the Statistical analysis section talks about regression-based model. They are not exactly the same, model for power analysis and primary analysis model should be as close possible as otherwise power analysis will be not be that meaningful.

Reviewer #2: Thank you very much for the invitation to review the manuscript by Brady and Ryan.

I read the study protocol with interest. The topic is certainly of interest and deserves closer attention.

This is clearly a complex intervention and therefore I wonder why the authors do not take the UK MRC framework for the development and evaluation of complex interventions into account. They do not report a programme theory or a logic model, do not consider a process evaluation and an economic evaluation. I wonder whether it is still timely to disregard the MRC framework, which has undoubtedly proven as golden standard.

Beyond this major comment, I have some minor comments, which might improve the manuscript.

The hypotheses passage is unnessessary and reads like a textbook information.

The sample size is rather low and I wonder whether the estimated effact size could be translated to a minimum clinical important difference. Apparently, a drop out rate has not been taken into account.

The authors do not completely adhere to the reporting statement. It remains unclear who will be in charge of the implementation of the allocation sequence and how this will be done.

Where will the intervention take place? Please add information.

Do the authors plan to implement a control mechanism whether blinding was successful?

A figure displaying the participant timelime is missing.

It remains unclear why an audit is not planned. Please comment on this. An audit would guarantee the quality of study procedures and data.

7. PLOS authors have the option to publish the peer review history of their article (what does this mean?). If published, this will include your full peer review and any attached files.

Reviewer #1: No

Reviewer #2: No

---

## [Author Response · Author response to Decision Letter 0]

27 Jun 2024

Reviewer Comments 

Author responses 

1. Why a medium effect size (d=0.5) is chosen, please explain the rationale behind it. 

As specified in the ‘trial design’ subheading of the ‘Materials and Methods’ heading, while ideally a large effect size (d=0.2) is preferable, this would equate to a sample size of 199. It will be expected there will be challenges in the recruitment of 199 participants for the scale and timeframe of this study (approx. 1 year), and due to the challenges noted in recruiting this population in the literature (Langbaum et al., 2023). Therefore, for pragmatic reasons, recruitment will aim for the sample size of 34 (d=0.5). 

2. Two primary outcome is chosen BADL and IADL. This will require multiple testing adjustment s otherwise type-1 error will be inflated. This is not addressed. 

Thank you for your comment, this is an error on my part. I have adjusted the primary outcome to be ADL performance as opposed to both BADL and IADL, as multiple testing adjustment is not planned. 

3. How the missing data will be handled? 

It is possible that some data will be missing. To avoid this, attempts to follow-up with all participants will be made. However, if missing data remain, intention-to-treat analysis will be used to account for this. 

4. The power analysis is based on two-sample t-tests, but the Statistical analysis section talks about regression-based model. They are not exactly the same, model for power analysis and primary analysis model should be as close possible as otherwise power analysis will be not be that meaningful. 

Thank you for this advice, appropriate revisions have been made in text. Types of tests used in analysis will be chosen following completion of the dataset to assess whether the data is parametric or non-parametric. 

This is clearly a complex intervention and therefore I wonder why the authors do not take the UK MRC framework for the development and evaluation of complex interventions into account. They do not report a programme theory or a logic model, do not consider a process evaluation and an economic evaluation. I wonder whether it is still timely to disregard the MRC framework, which has undoubtedly proven as golden standard. 

Discussion of the MRC framework has been applied to the development of CS-ADL. A process evaluation and an economic evaluation are outside the scope of this study and will be focused on in future research. 

The hypotheses passage is unnessessary and reads like a textbook information. 

Thank you for your recommendation, this section has been removed from the manuscript. 

The sample size is rather low and I wonder whether the estimated effact size could be translated to a minimum clinical important difference. Apparently, a drop out rate has not been taken into account. 

Thank you for this suggestion. In line with this suggestion, it has been included in text that the minimum clinically important difference will be calculated based on the medium effect size, by dividing the difference in scores from baseline to posttreatment by the standard deviation of baseline scores (Franceschini et al. 2023). 

In relation to the consideration of a drop-out rate, the following has been inserted in text: While this study is of a short duration, it is anticipated that there will be a minimum 10% attrition rate. This figure stems from Ritchie, Gillen & Grill’s (2023) analysis of dementia trial retention data. Therefore, recruitment will aim to exceed the estimated sample size by 10% to ensure the study is adequately powered. 

The authors do not completely adhere to the reporting statement. It remains unclear who will be in charge of the implementation of the allocation sequence and how this will be done. 

Random allocation will be completed by one of the principal investigator’s academic supervisors who are otherwise uninvolved in the conduct of the study 

Where will the intervention take place? Please add information. 

Intervention will take place at the clinical sites where recruitment will take place. CS-ADL groups will be facilitated in available group therapy/ occupational therapy rooms available. 

Do the authors plan to implement a control mechanism whether blinding was successful? 

The success of blinding will be tested following completion of outcome measures by asking assessors to guess the treatment assignment of participants, expressing either a treatment guess or uncertainty. If blinding is successful, ‘do not know’ guesses should be high, and incorrect guesses should be balanced (Kolahi, Bang & Park, 2009). While it is possible that unblinding might occur, i.e., outcome assessors become aware of group allocation, the process of blinding will be explained to participants, and they will be asked to not reveal their treatment allocation to the assessor. 

A figure displaying the participant timelime is missing. 

Apologies for this error. The figure in question has been uploaded with this resubmission. 

It remains unclear why an audit is not planned. Please comment on this. An audit would guarantee the quality of study procedures and data. 

While it is acknowledged that an audit would ensure the quality of study processes, traditional auditing methods are expensive and due to constraints on budget and resources for this study, a formal audit process will not be completed. The researcher will instead attend carefully to the research design throughout the process of the study. Furthermore, careful thought and consideration has gone into the planning of the pragmatics of the study to ensure processes are feasible within the clinical settings in which the study is run, to minimise deviations from the protocol.

---

## [Decision Letter · Decision Letter 1]

12 Aug 2024

Cognitive stimulation in activities of daily living for individuals with mild-to-moderate dementia (CS-ADL): study protocol for a randomised controlled trial.

PONE-D-23-41253R1

Dear Dr. Ryan,

We’re pleased to inform you that your manuscript has been judged scientifically suitable for publication and will be formally accepted for publication once it meets all outstanding technical requirements.

Kind regards,

Sascha Köpke

Academic Editor

PLOS ONE

Additional Editor Comments (optional):

Reviewers' comments:

Reviewer's Responses to Questions

**Comments to the Author**

1. Does the manuscript provide a valid rationale for the proposed study, with clearly identified and justified research questions?

Reviewer #2: Yes

2. Is the protocol technically sound and planned in a manner that will lead to a meaningful outcome and allow testing the stated hypotheses?

Reviewer #2: Yes

3. Is the methodology feasible and described in sufficient detail to allow the work to be replicable?

Reviewer #2: Yes

4. Have the authors described where all data underlying the findings will be made available when the study is complete?

Reviewer #2: Yes

5. Is the manuscript presented in an intelligible fashion and written in standard English?

Reviewer #2: Yes

6. Review Comments to the Author

You may also provide optional suggestions and comments to authors that they might find helpful in planning their study.

Reviewer #2: Thank you for the revision.

By adding methodological details about the randomisation and the UK MRC Framework, the transparency of the manuscript has improved very much.

7. PLOS authors have the option to publish the peer review history of their article (what does this mean?). If published, this will include your full peer review and any attached files.

Reviewer #2: No

---

## [Editor Report · Acceptance letter]

22 Aug 2024

PONE-D-23-41253R1 

PLOS ONE

Dear Dr. Ryan, 

I'm pleased to inform you that your manuscript has been deemed suitable for publication in PLOS ONE. Congratulations! Your manuscript is now being handed over to our production team.

Kind regards, 

on behalf of

Professor Sascha Köpke 

Academic Editor

PLOS ONE